# A Hierarchical Reinforcement Learning Based Optimization Framework for Large Scale Storage Location Assignment Problem

## Abstract

The Storage Location Assignment Problem(SLAP) is one of the essential problems within the domain of logistics. The objective is to dynamically allocate optimal storage locations to incoming items, aiming to maximize warehouse space utilization and operational efficiency. Prior research primarily focused on offline scenarios with predetermined goods arrival times. A smaller portion explored real-time allocation using heuristic algorithms based on manual rules and search methods. However, these methods suffer from inadequate solution quality and efficiency, particularly for large-scale problems. We draw inspiration from the partitioned, multi-layered, and modularized layout commonly adopted in most large-scale storage spaces to overcome this limitation. Building upon this inspiration, we propose a novel hierarchical optimization framework to solve large-scale SLAPs better via reinforcement learning. Specifically, we designed a two-level model: (1) a higher-level model learns to determine which block to choose, and (2) a lower-level model learns to select the final storage location under the constraints of the selected blocks in the upper level. We have designed a policy network based on attention mechanisms for SLAP to achieve better performance. To verify the effectiveness of the proposed framework, we collected a large amount of real historical data from the terminal operating system of Ningbo-Zhoushan Port and built a realistic container terminal simulator. Besides, we conducted extensive offline simulations and online testing using the simulator based on real data and validated the superior performance of our framework compared to existing benchmark methods.

## 1 Introduction

The Storage Location Assignment Problem (SLAP) concerns the allocation of products into a storage space and the optimization of the material handling costs or storage space utilization. As a standing and fundamental problem, SLAP can be applied in a wide range of logistical scenarios where a group requires temporary occupancy of a warehouse storage space, such as raw material warehouses, distribution centers, ports, and parking lots. Warehouse operations systems typically follow a sequential process, as illustrated in Figure 1, which includes the steps of reception, storage, order picking, and dispatch. In addition, SLAP, as an operational decision, plays a crucial role in optimizing the accommodation and picking process, affecting aspects such as batch definition, classification, routing, and order sequencing. The objective of SLAP is to minimize storage and costs while maximizing warehouse space utilization and operational efficiency.

SLAPs can be exceedingly difficult to solve as they are NP-hard combinatorial optimization problems. The main challenge of SLAP is the uncertainty of item arrival times, which makes it impossible to make arrangements in advance in an offline manner. Besides, SLAP involves several complex constraints and considerations that can be divided into five categories: capacity and conditions of the warehouse, characteristics of the products, configuration of the operation, and market and logistics resources. The most crucial category to consider is the capacity and physical conditions of the warehouse, which are related to the physical dimensions of the storage locations and the distribution of the storage area layout.

Various manual strategies such as "shortest path," "minimum centroid," and "first-in-first-out" have been proposed, showing promising results in simple scenarios. These strategies are ineffective in accommodating the complexity and uncertainties of real warehouse operations. Moreover, certain strategies that prioritize local optimizations may neglect the interconnectedness with other parameters. Furthermore, numerous studies have utilized traditional operations research (OR) or meta-heuristic methods, such as intelligent optimization algorithms, to address SLAPs. These methods are typically associated with high computational costs, often resulting in suboptimal solutions within given time constraints. Additionally, their design heavily relies on intricate domain knowledge. Recently, learning-based methods have shown the potential of high computational efficiency and competitive solutions. These studies primarily focus on small-scale SLAPs, considering fewer constraints than the large-scale SLAPs discussed in this paper. However, learning-based methods also have demonstrated potential in large-scale SLAPs with superior performance.

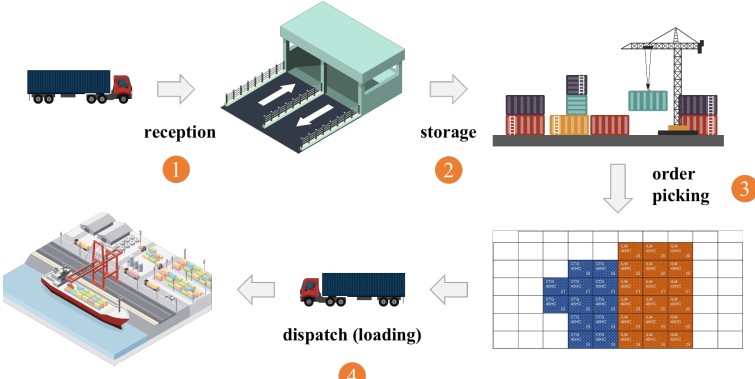

Figure 1: Warehouse operations systems are configured through the following sequential processes: (1) Reception: Receiving goods and materials into the warehouse from suppliers or others. (2) Storage: Organizing and placing the goods and materials into appropriate locations within the warehouse optimize space and improve efficiency. (3) Order picking: Retrieving requested goods and materials from their storage locations in the warehouse to fulfill orders. (4) Dispatch: Preparing and shipping the completed orders to their designated destination.

In this paper, we propose a novel optimization framework based on hierarchical reinforcement learning, namely H-SLAP, for solving large-scale SLAPs. Considering most large-scale storage spaces, e.g., three-dimensional warehouses adopt a partitioned, multi-layered, and modularized layout, we decompose SLAP into several sub-problems. We model the SLAP as an item-select problem, which contains two reinforcement learning (RL) agents. The higher-level agent functions as a block selector, responsible for choosing a block based on the distribution of stored goods. In contrast, the lower-level agent works as a location selector, which selects a specific storage location under the block constraints given by the block selector. We design a novel set of rewards to evaluate the rationality of storage positions in this task. To verify the effectiveness of the framework, we conducted offline and online evaluation experiments on a real port, which is a typical warehouse with a three-dimensional. We collected real historical data from the terminal operating system of Ningbo-Zhoushan Port and built a realistic container terminal simulator. The main contributions are concluded as follows:

- We systematically explore the storage location assignment task, and propose a novel hierarchical reinforcement learning framework. To the best of our knowledge, we are the first to bring a practical hierarchical reinforcement learning framework for the storage location assignment problem.
- To verify the effectiveness of the framework proposed in this paper, we constructed a simulation system using real port data and open-sourced it here for interested researchers to use.
- We designed a policy network based on an attention mechanism that effectively utilizes the information relationships among items, resulting in impressive performance in large-scale storage location assignment tasks.
- We conducted extensive offline and online evaluation experiments, model analyses, and ablation tests, demonstrating that H-SLAP significantly improves optimization objectives compared to

existing methods. Additionally, the hierarchical framework reduces the computational complexity from the original M*N to M+N, thereby enhancing training efficiency.

## 2 RELATED WORK

**Storage Location Assignment:** The Storage Location Assignment Problem (SLAP) involves allocating products to storage spaces (Reyes et al. (2019)). In terms of complexity, (Frazelle (1989)) classifies SLAP as NP-Hard, considering the variations caused by the number of products and warehouse storage characteristics. Traditional approaches primarily rely on various optimization methods, such as mixed-integer programming models (Yang et al. (2015)), heuristic algorithms and procedures (Battini et al. (2015)), meta-heuristics like taboo search (Zhang et al. (2021) Kübler et al. (2020) Yang et al. (2021) Otto et al. (2017)), simulation based on discrete events (Pan et al. (2012)), class-based policies and rules (Ene et al. (2016)), and multi-criteria methods like Electre III (Fontana & Nepomuceno (2017)).

**Learning-based Method for SLAP:** Recent research has explored the feasibility of using Deep Reinforcement Learning (DRL) for various variations of the dynamic storage location assignment problem. (Kim et al. (2020)) addressed Dynamic SLAP(DSLAP) in a ship block stockyard, aiming to minimize block rearrangement. (Waubert de Puiseau et al. (2022)) studied a different DSLAP variation where a DRL agent assigned pallets to zones upon arrival. Both studies reported significant improvements with DRL compared to existing methods.

**RL for Combinatorial Optimization:** Numerous studies have demonstrated the effectiveness of RL in COPs. For instance, (Cappart et al. (2021)) combined RL and constraint programming methods to address the Traveling Salesman Problem, exhibiting favorable performance in terms of solution quality and computation time. Additionally, RL-based approaches also contribute to solving the problem of medical resource allocation(Hao et al. (2021)), hybrid flow shop scheduling(Ni et al. (2021)), the three-dimensional bin packing problem (Zhao et al. (2021)), facility location problems(Wang et al. (2023)), electricity pricing(Chung et al. (2020)) and urban subway network expansion(Wei et al. (2020)). By harnessing the learning and adaptive capabilities of RL algorithms, optimal solutions can be discovered in intricate and evolving problem domains.

## 3 PROBLEM FORMULATION

In this paper, we focus on the SLAP in three-dimensional warehouses, which typically consist of racks, stacker cranes, transportation equipment (including automated vehicles), and loading/unloading platforms. In our case, we have chosen the port as the scenario for modeling. For the containers arriving at the port in real time, it is necessary to arrange appropriate and reasonable storage spaces to improve the efficiency of subsequent loading and shipping operations.

In a container terminal yard with hundreds of thousands of storage positions, each storage position has an identification number id represented as $(a, b, r, t)$, where $a$ denotes the block number, and $(b, r, t)$ represent the bay number, row number, and tier number respectively. In practical applications, a feasible storage position should satisfy the following constraints:

- **Prohibition of Suspension:** Containers must not be suspended in mid-air.
- **Height Restriction:** It is not allowed to exceed the maximum stacking height $L_{tier}$ specified for the storage area.
- **Cargo Reshuffling**: A certain number of empty slots shall be reserved in each bay for cargo reshuffling.
- **Size Constraint:** Only containers of the same dimensions can be stacked together in the same stack.

Based on the principle of the highest efficiency, stacker load balancing principle and transportation equipment path optimization principle, the storage layout of n container storage locations corresponding to a ship can be quantitatively measured by the following five indicators, Detailed indicators are shown in Appendix A.1:

$$R = \sum_{i=1}^{5} (\omega_i * r_i), \quad r = \{r^{equilibrium}, r^{distance}, r^{block\_num}, r^{reshuffle}, r^{concentration}\} \quad (1)$$

In addition, the main challenge of SLAP in a terminal's container yard is that the sequence of incoming containers is uncertain, preventing the pre-planning of suitable container positions. Real-time algorithms based on models have low execution efficiency and, with the increase of the number of containers and the scale of the container yard, the complexity of obtaining an optimal solution increases exponentially. It is difficult to obtain a good solution within an acceptable time frame.

# 4 METHODOLOGY

## 4.1 OVERALL FRAMEWORK

In this paper, we propose a novel hierarchical reinforcement learning-based optimization framework, consisting of two steps, as shown in Figure 2. In Step 1, after obtaining the information on ship berth allocation, we utilize a simulated annealing algorithm (Wang et al. (2022)) for coarse-grained container yard pre-planning order to reduce the number of available slots. The detailed algorithm flow can be found in Appendix A.3. Due to the large number of feasible storage positions (up to tens of thousands) after pre-planning, the large discrete action space makes it difficult to apply reinforcement learning methods. Therefore, in Step 2, we design two-level agents, with information available in the subsequent chapters. Section 4.2 introduces the reinforcement learning formulation and policy network architecture, while Section 4.3 presents the hierarchical training algorithm.

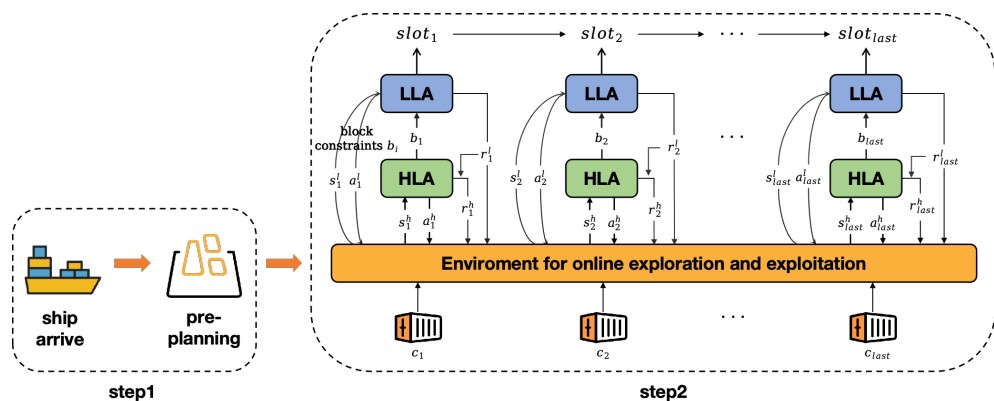

Figure 2: Hierarchical Optimization Framework

## 4.2 HIERARCHICAL REINFORCEMENT LEARNING

### 4.2.1 RL FORMULATION

In this section, we formulate the Block Select Process and Slot Select Process as MDPs separately. The elements of this MDP are as follows:

**A. High Level: Block Selector**

State $s_t^h$: $s_t^h$ consists of n blocks' feature and each block feature contains the distance between the block and the berth, the number of containers on the same route in the block, the total number of containers/block capacity in the block, the number of empty bays (20TEU), the remaining capacity of 20TEU container spaces, the remaining capacity of 40TEU container spaces, and the quantity of containers of various weight grades (The containers here refer to the containers that have the same route as the containers to be allocated at the position.) in the block.

Action $a_t^h$: The action $a_t^h$ is defined as the block $b_i \in B$ selected at step $t$.

Reward Function $r(s_t^h, a_t^h)$: The high-level reward $r^h$ is measured by block-equilibrium, block-berth distance, and Block Num(Zhen et al. (2022) Feng et al. (2022)). Due to the sparse nature of the rewards and the fact that accurate values are only obtained after all containers have been stored in the yard, we specifically designed some immediate reward functions (see Appendix A.2) to ensure that the agent can quickly learn and update its strategies. The high-level agent also receives rewards

from the lower-level agent. It can adjust its policy-making strategy based on the performance and reward information from the lower-level agent.

**B. Low Level: Slot Selector**

State $s_t^l$: $s_t^l$ consists of n slots' feature and each slot feature contains the information about the incoming container, the bay-row-tier information of the slot, the number of containers the column, the number of containers on the same route in the stack, the weight of containers in each tier, and whether they are on the same route.

Action $a_t^l$: The action $a_t^l$ is defined as the slot $s_i \in b_i$ selected at step $t$. Here, $b_i$ is the block of action selection at the high level.

Reward Function $r(s_t^l, a_t^l)$: The low-level reward $r_l$ is measured by reshuffle rate and concentration. Just like the High-Level Reward Function, we have also designed some immediate reward functions (See Appendix A.2) to help the Low-Level agent quickly learn the optimal strategy.

### 4.2.2 POLICY NETWORK ARCHITECTURE

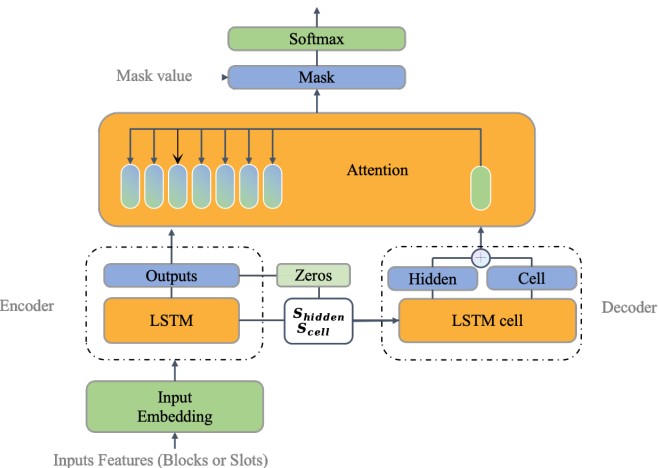

Figure 3: Policy network. It takes the block/slot state st as input and generates the probability distribution to select a block/slot.

We use a sequence-to-sequence framework similar to the Pointer Network (Vinyals et al. (2015)) as our Policy Network. The advantage of this framework is that the model's output points to the input content. Our input consists of variable-length sequences of block/slot features, and the output is an index pointing to the input sequence. The specific structure is as follows:

**Embedding:** Firstly, the input sequence features are encoded through Embedding and fed into the Encoder.

**Encoder:** We choose to use an LSTM as the encoder. The embedded sequence is input to the encoder, which produces the hidden layer Encoder outputs, hidden state vector, and cell state vector.

**Decoder:** We choose to use an LSTM(Shi et al. (2015)) unit as the decoder. The zero vector with the same dimension as the Encoder outputs, along with the hidden state and cell state, is inputted into the decoder. The output is a new hidden state vector and cell state vector, which are summed and inputted into the Attention layer.

**Attention:** In this layer, we perform attention calculation using the hidden layer of the Encoder and the hidden state vector of the Decoder, as shown in Equation 2. We directly use the attention weight information as the probability distribution of positional importance.

$$u_j^i = v^T tanh(W_1 e_j + W_2 d_i) \qquad j \in (1, ..., n) \tag{2}$$

Here, $e_j$ represents the hidden layer output of the encoder at time step j, and $d_i$ represents the hidden state output of the decoder at time step i. $W_1$, $W_2$, $v^T$ are training parameters.

**Mask:** To allow the model to have a better understanding of the placement situation of the decoding head, certain illegal placements in the input sequence cannot be chosen. Therefore, we use a masking approach where we replace the non-selectable positions with $-\infty$ based on the input masking sequence:

$$u_{mask} = \begin{cases} u_i, & mask_i = 0 \\ -\infty, & mask_i \neq 0 \end{cases} \tag{3}$$

**Softmax:** By applying softmax to the masked position vector $u^i$ (Equation 4, we obtain the probability distribution of the masked positional importance. By applying softmax to the masked position vector $u$ (Equation 4), we obtain the probability distribution of the masked positional importance.

$$p(C, P) = softmax(u_{mask}) \tag{4}$$

### 4.3 SOFT ACTOR-CRITIC TRAINING

In this section, our objective is to train the hierarchical policy network with network parameters $\theta_1$ and $\theta_2$ to maximize the expected cumulative rewards in equations (6) and (7). Both the high-level and lower-level policy networks are trained using the Soft Actor Critic (SAC) algorithm, in which the actor is synonymous with our policy network. In the high level, the actor outputs a probability distribution over actions (i.e., blocks), while in the lower level, the actor outputs a probability distribution over actions (i.e., slots). The critic is responsible for evaluating the expected cumulative rewards of allocating subsequent containers to block-level and slot-level, respectively. This evaluation helps to reduce training variance.

When incoming containers arrive, the high-level agent obtains the state $s_t^h$, mask $m_t^h$, action $a_t^h$, and reward $r_t^h$ according to 4.3.1 A. Under the constraint of the high-level action $a_t^h$, the lower-level agent obtains the state $s_t^l$, mask $m_t^l$, action $a_t^l$, and reward $r_t^l$ according to Section 4.3.1 B. After incoming containers complete their action, we obtain the new states $s_{t+1}^h$ and $s_{t+1}^l$, and two new transitions $e_t^h = (s_t^h, m_t^h, a_t^h, r_t^h, s_{t+1}^h)$ and $e_t^l = (s_t^l, m_t^l, a_t^l, r_t^l, s_{t+1}^l)$ are generated and saved to the replay buffer. It is worth noting that in the lower level, we adopt a similar approach to distributed reinforcement learning, where the agents in each block share a set of parameters $\theta$ and maintain their transitions, that is, in the transition corresponding to the last st+1 when there are no more incoming containers in a certain block, done is set to True. We store the transitions into buffer $D^h = \{e_{t1}^h, e_{t2}^h, \ldots\}$ and $D^l = \{e_{t1}^l, e_{t2}^l, \ldots\}$ during the running of simulator. During the training, we apply Q-learning updates on uniformly sampled transitions $(s, m, a, r, s') \sim U(D)$ from the replay buffer. The model updates at iteration l uses the following loss function:

$$L(\phi_i, D) = E_{(s,m,a,r,s',d) \sim D} \left[ (Q_{\phi_i}(s, a) - y(r, s', m', d))^2 \right] \tag{5}$$

where the target is given by:

$$y(r, s', m', d) = r + \gamma(1 - d) \left( \min_{k=1,2} Q_{\phi_{\text{targ},ik}}(s', \tilde{a}') - \alpha \log \pi_{\theta_i}(\tilde{a}' \mid s', m') \right),$$
$$\bar{a}' \sim \pi_{\theta_i}(\cdot \mid s', m') \tag{6}$$

The training details are shown in Algorithm 2 in Appendix A.4.

# 5 EXPERIMENTS

## 5.1 EXPERIMENTS SETTINGS

We start by designing a simulator to investigate the contributions of the proposed framework. The details of the simulator can be found in Appendix B. To ensure a better distinction in performance across different-sized routes, we considered two scales of export container route data: 572 containers and 1000 containers. The dataset for 572 containers consists of 13 container zones with nearly 10,000 container slots, while the dataset for 1000 containers includes 20 container zones with approximately 20,000 container slots. During the training process, we shuffle the order of container arrivals to better simulate the randomness and variability of actual operations. In the testing phase, we use a test set that is shuffled in a manner consistent with the size of the training set. More detailed information about the dataset can be found in the appendix.

The experimental settings for the baselines and our method are as follows: we first train the models on each type of training dataset and then evaluate them on corresponding test datasets of the same scale. For single-level methods, containers are directly selected from all available container slots. In the case of layered methods, the high-level agent selects the container zone, and then the lower-level agent chooses the container slot within the selected zone. Both the higher-level and lower-level agents are trained simultaneously. For other methods, such as expert policies and random methods, we directly evaluate them on different test datasets of varying scales. The experimental results are obtained by running each algorithm ten times to obtain the mean and variance values of the optimization objective.

## 5.2 BASELINES

We implement several competitive models as baselines. The implementation details of each method are in Appendix D.

**Random:** Randomly assigns slot positions from the available options, and serves as a blank control group.

**Expert Strategy1:** We use the immediate reward defined in Appendix A.2 as a significant criterion for sorting, whereby storage locations with higher immediate rewards are given absolute priority.

**Expert Strategy2:** We define four different priority criteria for stack selection and randomly choose one slot with the highest priority. These four priorities in descending order are stack with the same attribute on the top container, stack with lighter weight on the top container, empty stack, and other stacks.

**Expert Strategy3:** We address this problem in two steps: First, we select the block with the fewest number of containers with the same vessel. Second, we define four different priority criteria and randomly choose one slot with the highest priority in the selected block. These four priorities in descending order are stack with containers of the same attributes, stack with lighter containers, empty stack, and other stacks.

**Single-SAC (Christodoulou (2019)):** A non-hierarchical reinforcement learning method, which the policy network from Chapter 4.2.2 is utilized to select the with the highest score, and the training framework applies SAC.

## 5.3 COMPARISON WITH BASELINES

We evaluated the trained models and expert policies based on simulation environment and used multiple tests to reduce the impact of randomness. The complete results are shown in Appendix E.1 Table 5. Tables 1 show that our method consistently outperforms all baselines on all datasets (higher overall objectives are better). On some datasets, the expert policy performs extremely poorly, which is because greedy strategies, such as not considering the future placement of boxes, often lead to local optima, improving only box flipping without improving concentration or vice versa., Table 1 shows that Single-SAC performs much worse than the expert policy, due to the large action space in large-scale storage space scenarios, making it difficult to explore positive training trajectories. In contrast, our hierarchical framework greatly reduces the action space, and the high-level agents

can allocate the internal states of the box area, while the lower-level agents can place the boxes in the appropriate positions and consider future situations better. The cooperation between these two levels of agents also makes our method better in future planning.

As an example, we visualized the results of storage location allocation for 572 containers in Experiment 1 (Figure 4. Figure 4(a) illustrates the distribution at the block level, while Figure 4(b) shows the distribution within each block. More results can be found in the appendix E.2.

| Experiment | Method | Block-equilibrium ↓ | Block Num ↓ | Reshuffle rate ↓ | Concentration ↓ | Yard-berth distance ↑ | Objective ↑ |
|---|---|---|---|---|---|---|---|
| | Random | 0.3482 | 5 | 0.1204 | 0.1114 | 0.9725 | -59.3828 |
| | Expert Strategy1 | 0.0369 | 0 | 0.1273 | -0.0328 | 1.0 | 36.8411 |
| | Expert Strategy2 | 0.5933 | 0 | 1.6945 | **-0.0650** | 0.875 | -178.5391 |
| 572-1 | Expert Strategy3 | **0.0032** | 5 | 1.7382 | -0.0368 | 0.9709 | -171.9125 |
| | Single-SAC | 0.3549 | 5 | 0.0471 | 0.0471 | 0.9803 | -112.3187 |
| | Hierarchical-SAC-MLP | 0.5776 | 0 | 1.6701 | -0.0554 | 1.0 | -169.2379 |
| | **Hierarchical-SAC(Ours)** | 0.0401 | **0** | **0.0418** | -0.0171 | **1.0** | **43.5078** |
| | Random | 0.3250 | 5 | 0.1003 | 0.1225 | 0.9824 | -55.6716 |
| | Expert Strategy1 | 0.2452 | 0 | 0.1772 | -0.0498 | 1.0 | 12.7313 |
| 300-1 | Expert Strategy2 | 0.2787 | 0 | 1.5518 | **-0.0598** | 1.0 | -127.0680 |
| | Expert Strategy3 | **0.0060** | 5 | 1.4481 | -0.0331 | 0.9753 | -143.3403 |
| | **Hierarchical-SAC(Ours)** | 0.0423 | **0** | **0.0969** | -0.0381 | **1.0** | **39.8773** |
| | Random | 0.4734 | 6 | 0.1859 | 0.0669 | 0.9512 | -85.0726 |
| | Expert Strategy1 | 0.0384 | 0 | 0.1689 | -0.0496 | **1.0** | 33.9667 |
| 1000-1 | Expert Strategy2 | 0.6298 | 0 | 2.2920 | **-0.0685** | 0.875 | -241.5824 |
| | Expert Strategy3 | **0.0081** | 6 | 1.6325 | -0.0460 | 0.9375 | -172.5890 |
| | **Hierarchical-SAC(Ours)** | 0.0619 | **0** | **0.0979** | -0.0131 | 0.9780 | **34.2254** |

Table 1: Experimental results under different datasets

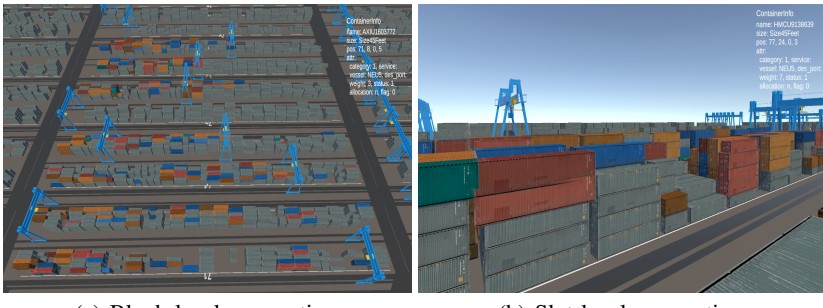

(a) Block-level perspective        (b) Slot-level perspective

Figure 4: Visualization results of dataset 572 (Hierarchical-SAC)

## 5.4 ABLATION STUDIES

**Effect of Two Levels of Model:** The results in Table 2 show that our proposed H-SLAP framework significantly reduces computational complexity and lowers the requirements for GPU memory compared to directly selecting storage positions from all possible positions. In a 16G GPU memory environment, the batch of a single-layer approach can reach a maximum of 2, while the batch size of a hierarchical approach can be set to a maximum of 32. Additionally, as shown in figure 5, reducing the action space through action hierarchy lowers the cost of exploration, enabling the hierarchical approach to converge more quickly to an optimal strategy compared to a single-layer approach.

| Method | cost time(s/episode) ↓ | batch size ↑ |
|---|---|---|
| Single-SAC | 1051.69 | 2 |
| **Hierarchical-SAC(Ours)** | 162.53 | 32 |

Table 2: Comparison of cost time between Hierarchical and Single SAC

**Effect of Our Policy Network:** We conducted ablation tests to validate the effectiveness of our designed policy network. Table 3 shows the results of various ablation settings. We found that attention and LSTM are indispensable. Here, we replaced LSTM and attention with fully connected layers

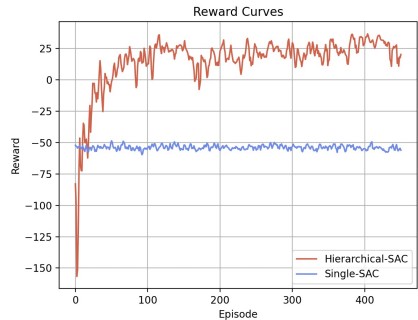

Figure 5: Comparison of convergence rewards between Hierarchical and Single SAC

and sigmoid as part of the ablation settings. The policy network utilizes the attention mechanism to learn the probability distribution of selecting positions, allowing the network to focus on different parts of the input sequence and make better choices.

| Task | Block-equilibrium ↓ | Block Num ↓ | Reshuffle rate ↓ | Concentration ↓ | Yard-berth distance ↑ | Objective ↑ |
|---|---|---|---|---|---|---|
| **Hierarchical-SAC(Ours)** | **0.0401** | **0** | **0.0418** | -0.0171 | **1.0** | **43.5078** |
| - attention | 0.2024 | 0 | 0.1465 | -0.0258 | 1.0 | 17.6788 |
| - LSTM & attention | 0.5776 | 0 | 1.6701 | **-0.0554** | 1.0 | -169.2379 |

Table 3: Ablation tests on dataset 572

**Generalization:** To validate the generalizability of our method, we tested the lower-level model trained on a dataset of 572 containers on a larger-scale dataset. As shown in Table 4, the lower-level model trained on 572 containers exhibited similar performance to the models trained on datasets of 300 and 1000 containers. This indicates that our method can quickly generalize to new problems of varying scales without the need for retraining.

| Dataset | Model | Block-equilibrium ↓ | Block Num ↓ | Reshuffle rate ↓ | Concentration ↓ | Yard-berth distance ↑ | Objective ↑ |
|---|---|---|---|---|---|---|---|
| 300-1 | Trained on 300 | 0.0423 | 0 | 0.0969 | -0.0381 | 1.0 | 39.8773 |
| | Trained on 572 | 0.0891 | 0 | 0.0903 | -0.0364 | 1.0 | 35.6967 |
| 300-2 | Trained on 300 | 0.0490 | 0 | 0.1070 | -0.0414 | 1.0 | 38.5395 |
| | Trained on 572 | 0.0824 | 0 | 0.0668 | -0.0364 | 1.0 | 38.7068 |
| 1000-1 | Trained on 1000 | 0.0619 | 0 | 0.0979 | -0.0131 | 0.9780 | 34.2254 |
| | Trained on 572 | 0.0619 | 0 | 0.0880 | -0.0216 | 0.9779 | 36.0557 |
| 1000-2 | Trained on 1000 | 0.0614 | 0 | 0.0853 | -0.0099 | 0.9779 | 35.2111 |
| | Trained on 572 | 0.0589 | 0 | 0.0952 | -0.0211 | 0.9777 | 35.5832 |

Table 4: Generalization verification

## 6 CONCLUSION

In this paper, we investigate the problem of storage location assignment and model it as an item selection problem. We propose a novel hierarchical reinforcement learning method to learn the location selection policy. We utilize a container storage allocation simulator based on real port data and conduct extensive experiments under various settings. The results demonstrate that our method consistently outperforms the baseline methods. Furthermore, compared to other reinforcement learning-based approaches, our proposed method achieves better performance and convergence while significantly reducing computational complexity.

There are several future directions for our work. Firstly, we currently adopt a first come, first served strategy for allocation. In the future, we plan to expand our approach by temporarily storing containers in a shared waiting buffer before allocating storage locations as a group. This will help to reduce performance loss caused by future uncertainty. Secondly, we will validate our proposed framework in more warehouse environments in the future.

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

# A  APPENDIX

## A.1  DETAIL OF INDICATORS

**Notation**

| | |
|---|---|
| $i, j$ | Serial number of container |
| $C_{p,w}^{s,v}$ | A set of containers with service $s$, vessel $v$, port of discharge $p$, weight class $w$ |
| $A_v$ | A set of blocks where there are containers with vessel v |
| $n\_A_{s,v}$ | length of $A_{s,v}$ |
| $b, r, t$ | Serial number of bay, stack, tier |
| $d_{a,b}^{ber}$ | Distance between bay $(a, b)$ and berth ber |
| $d_a^{ber}$ | the distance between $block_a$ in $berth_{ber}$ |
| $L_a$ | Limits tiers in block $a$ |
| $T_{a,b,r}$ | The number of container types in the stack $(a, b, r)$ |
| $NT_{a,b,r}$ | The maximum number of containers of a certain type in the stack $(a, b, r)$ |
| $\sum_{a,b,r,t} x_{b,r,t}^{i,a}$ | the total number of containers in the yard |
| $x_{b,r,t}^{i,a}$ | $x_{b,r,t}^{i,a}$=1 if $container_i$ in slot $(a, b, r, t)$ and $container_i$ is not belong to init containers, else 0 |
| $ub_s^a$ | the expected quantity of $vessel_s$ in $block_a$ |
| $ub_s$ | the expected quantity of blocks for $vessel_s$ |
| $ref_i$ | the reshuffle num after slot $container_i$ |
| $k_s^a$ | $k_s^a$=1 if container of $vessel_s$ slot in $block_a$ else 0 |
| $N_s^{block\_limit}$ | the limit of Block Num for vessel s |

The final metrics are presented as follows, where the specific symbol meanings are shown in Table A.1.

- **Block Equilibrium:** During the loading process, multiple container areas will simultaneously carry out the container lifting operation to balance the workload among different areas $\{N_0, N_1, ..., N_i\}$. This practice is advantageous in terms of improving equipment utilization and preventing congestion.

$$\sum_{a \in A_v} | \sum_{b,r,t} \sum_{i \in C^v} x_{b,r,t}^{i,a} - \frac{\sum_{a \in A_v} \sum_{b,r,t} \sum_{i \in C^v} x_{b,r,t}^{i,a}}{n\_A_v} | \tag{7}$$

- **Yard-berth Distance:** According to the order of berths, each berth corresponds to a specific area. Stacking containers in the designated based on berths can improve management, efficiency, and safety.

$$1 - \frac{\sum_{a,b,r,t,i} d_{a,b}^{ber} * x_{b,r,t}^{i,a}}{max(d_{a,b}^{ber}) * \sum_{a,b,r,t} x_{b,r,t}^{i,a}} \tag{8}$$

- **Block Num:** The distribution of export containers on a vessel among different container areas depends on the total container volume $\sum(v)$, as scattered storage locations can negatively impact management and container lifting operations.

$$\begin{cases} n\_A_v - N_s^{block\_limit} & n\_A_v > N_s^{block\_limit} \\ 0 & otherwise \end{cases} \tag{9}$$

Here, $(t_2 - t_1)$ means the level gap between $con_i$ and $con_j$, and $(y_{a,b,r}^{i,t_1,j,t_2} - z_{a,b,r}^{i,t_1,j,t_2}) = 1$ means the position relationship of any pair of containers in the same stack does not conform to the

priority constraint. $y_{a,b,r}^{i,t_1,j,t_2} = 1$ if $container_i$ in $slot(a,b,r,t_1)$, $container_j$ in $slot(a,b,r,t_2)$ else equals 0. $z_{a,b,r}^{i,t_1,j,t_2} = 1$ if $y_{a,b,r}^{i,t_1,j,t_2} = 1$, and the position relation between $con_i$ and $con_j$ does not conform to weight constraint, otherwise equals 0.

- **Reshuffle Rate:** After the given stowage plan for a vessel, a fixed sequence of container lifting in the container yard $S = \{C_0, C_1, ..., C_n\}$ will be generated. If the sequence of lifting containers from top to bottom does not match the sequence S, re-stowage is necessary to extract containers of lower levels but with an earlier sequence. At this time, the container lifting equipment may become invalid and therefore have to wait or conduct re-stowage operations first.

$$\frac{\sum_{a,b,r,i,j,t_i < t_2} (y_{a,b,r}^{i,t_1,j,t_2} - z_{a,b,r}^{i,t_1,j,t_2}) * (t_2 - t_1)}{\sum_{a,b,r,t} x_{b,r,t}^{i,a}} \tag{10}$$

- **Concentration:** The concentration considers the number of types of containers in each column $T\{a,b,r\}$ as well as the number of each type $len(T\{a,b,r\})$. Ideally, the attributes of containers in each column should be exactly the same. During loading, the containers with the same attributes have adjacent lifting order and can be interchanged, which can effectively reduce restowage and limit the back-and-forth movement of container lifting, thus improving concentration.

$$\frac{\sum_{a,b,r}(k_1 * \frac{T_{a,b,r} - 1}{\sum_t x_{b,r,t}^{i,a}} + k_2 * (1 - \frac{NT_{a,b,r}}{L_a}))}{\sum_{a,b,r,t} x_{b,r,t}^{i,a}} \tag{11}$$

## A.2 Detail of Immediate Reward Function

- **Block Equilibrium:**

$$r_{blockEquilibrium}^{i,a} = -1 * ln(1 + max(0, \sum_{b,r,t,i} x_{b,r,t}^{i,a} - ub_s^a)) \tag{12}$$

- **Yard-berth Distance:**

$$r_{berth-block}^{ber,a} = \begin{cases} 0.5 & \text{if } d_a^{ber} = 0 \\ -1 * d_a^{ber} \end{cases} \tag{13}$$

- **Reshuffle Rate:**

$$r_{weightdiff}^i = \begin{cases} 0.5 & \text{if } ref_{i-1} = ref_i \\ ref_{i-1} - ref_i \end{cases} \tag{14}$$

- **Concentration:**

$$r_{concentration}^i = \begin{cases} -0.1 & \text{if } \sum_{r,t,i} x_{b,r,t}^{i,a} = 1 \\ \sum_t s_{b,r,t}^{i,a} \end{cases} \tag{15}$$

The definition of $s_{b,r,t}^{i,a}$ is as follows.

$$s_{b,r,t}^{i,a} = \begin{cases} 1 & \text{if } container_i \text{ in stack(a,b,r)}, container_j \text{ in stack(a,b,r) with same vessel} \\ -1 & \text{if } container_i \text{ in stack(a,b,r)}, container_j \text{ in stack(a,b,r) with different vessel} \end{cases} \tag{16}$$

- **Block Num:**

$$r_{blockoverflow}^i = \begin{cases} -1 * max(0, \sum_a k_s^a - ub_s) & \text{if container of } vessel_s \text{ slot in } block_a \text{ firstly} \\ 0 \end{cases} \tag{17}$$

## A.3 Detail of SA-InitialBlocks

In practical scenarios at traditional terminals, each shipping route is associated with a predefined set of container blocks. Containers for a specific route are typically stored within the corresponding

designated block range. This approach significantly reduces the action space for the higher-level agent's block selection.

To automate the selection of container blocks for each shipping route, we employ the simulated annealing (SA) algorithm instead of manual block range selection.

Given the remaining capacity of each container block and the number of containers for the target shipping routes, the SA algorithm allocates blocks to each route based on their capacity requirements. Additionally, to provide a reasonable exploration space for the higher-level agent, we introduce a multiplier by multiplying the number of containers for the target routes. This results in a larger block range, enhancing redundancy and facilitating effective exploration. During the allocation process, we consider three primary indicators: berth-block suitability, block concentration, and quantity balance. The SA algorithm follows a specific process, as presented in the Algorithm 1.

---

**algorithm 1** SA-InitialBlocks

---

**Input:** vessels $V$, the available slots for each container block $B\{b1, b2, ...b_s\}$. yard container blocks size $s$. Simulated annealing parameters end temperature $\tau_{end}$, cool rate $\varphi$, iterations of each temperature $\zeta_{end}$.

**Output:** Container allocation for each vessel in each container block C($c_2^1$ represents the number of containers allocated to vessel 1 in container block 2, corresponding to the number of available slots in each container block).

1: Generate a random solution by assigning random container quantities $C_b^v$ to each container block S for each vessel V. This solution is denoted as X.
2: $\tau := \tau_{start}, \zeta := 1.$
3: **while** $\tau > \tau_{end}$ **do**
4:     **while** $\zeta < \zeta_{end}$ **do**
5:         Randomly select $v, b, b2, c$.
6:         $C_b^v += c, C_{b2}^v -= c$. /* This generates a new neighbor solution $X_{new}$. */
7:         $\Delta Z = Z(X) - Z(X_{new})$. /* function Z calculates Berth-Block Correspondence, Block Concentration Container, and Quantity Balance of the solution */
8:         **if** $\Delta Z > 0$ **then**
9:             $X = X_{new}$ /* Accept the new solution*/
10:         **else**
11:             Accept $X_{new}$ with a probability $e^{\frac{\alpha\Delta Z}{\tau}}$
12:         **end if**
13:         $\zeta += 1, \tau* = \varphi$
14:     **end while**
15: **end while**
16: **return** $X_{best}$

---

### A.4 Detail of Hierarchical-SAC Training Process

Algorithm 2 shows the training process of hierarchical Soft Actor Critic.

## B Simulator

### B.1 Static Yard Environment Configuration

The terminal yard in the simulator replicates the layout of 68 container zones and 18 berths at Meishan Terminal in Ningbo-Zhoushan Port. The specifications, spatial relationships, and actual stacking areas of each container zone exactly match those of the real yard. To recreate the original storage conditions at a specific moment, we utilized historical records from the Ningbo-Zhoushan Port's Terminal Operating System (TOS) between June 2022 and December 2022.

### B.2 Main Features of the Simulator

The simulator can dynamically allocate available empty container slots that meet the requirements of Prohibition of Suspension, Height Restriction, Cargo Reshuffling, and Size Constraint for upcoming

---

**algorithm 2** Hierarchical reinforcement Learning Training with SAC(Discrete)

---

**Input:** Training episode M; Initial policy parameters $\theta_h$,$\theta_l$, Q-function parameters $\phi_{h0}, \phi_{h1}$, $\phi_{l0}, \phi_{l1}$.

**Output:** Trained discrete models

1: $\phi_{targ,h0} \leftarrow \phi_{h0}, \phi_{targ,h1} \leftarrow \phi_{h1}, \phi_{targ,l0} \leftarrow \phi_{l0}, \phi_{targ,l1} \leftarrow \phi_{l1}$     ▷ Equalise target and local network weights

2: $R_h, R_l \leftarrow \emptyset$                                                        ▷ Initialise empty replay buffers

3: **for** each iteration **do**

4:     **for** each environment step **do**

5:         $a_h \sim \pi_{\theta_h}(\cdot \mid s, m)$                          ▷ Sample action from the high level policy

6:         $a_l \sim \pi_{\theta_l}(\cdot \mid s_{a_h}, m_{a_h})$                   ▷ Sample action from the low level policy

7:         $s', m', s_b', m_b', r_h, r_l, d \sim p(s', s'_{a_h} \mid s, s_{a_h}, a_h, a_l)$                ▷ Get transition from the environment

8:         $R_h \leftarrow R_h \cup \{(s, m, a_h, r_h, s', m', d)\}$       ▷ Store the transition in the high level buffer

9:         $R_l \leftarrow R_l \cup \{(s_{a_h}, m_{a_h}, a_l, r_l, s'_{a_h}, m'_{a_h}, d)\}$▷ Store the transition in the low level buffer

10:     **end for**

11:     **for** i = high or low level **do**

12:         **for** each gradient step **do**

13:             $\phi_{ik} \leftarrow \phi_{ik} - \lambda_Q \hat{\nabla}_{\phi_{ik}} J(\phi_{ik})$ for $k \in \{0,1\}$     ▷ Update the Q-function parameters

14:             $\theta_i \leftarrow \theta_i - \lambda_\pi \hat{\nabla}_{\theta_i} J_\pi(\theta_i)$                        ▷ Update policy weights

15:             $\alpha_i \leftarrow \alpha_i - \lambda \hat{\nabla}_{\alpha_i} J(\alpha_i)$                            ▷ Update temperature

16:             $\phi_{targ,ik} \leftarrow \tau\phi_{targ,ik} + (1-\tau)\phi_{ik}$ for $k \in \{0,1\}$  ▷ Update target network weights

17:         **end for**

18:     **end for**

19: **end for**

20: **return** $\theta_h, \theta_l, \phi_{h0}, \phi_{h1}, \phi_{l0}, \phi_{l1}$

---

container arrivals. Upon selecting a slot for placement, the simulator updates the storage status accordingly. It can monitor the storage status of each position in the yard, and calculate real-time yard performance metrics, such as estimated workload in different container zones during vessel loading operations and the number of rehandles for each container stack, to provide instant feedback for algorithms.

### B.3 SIMULATOR ACCELERATION

Despite replicating a large number of container slots in the simulator, reaching tens of thousands, it achieves fast calculations at the millisecond level for tasks such as available slot screening, zone feature computation, slot feature computation, and yard stacking metric calculation. This efficiency is achieved through two aspects. Firstly, the simulator benefits from caching critical yard storage information. Secondly, during the container entry process, the simulator only performs real-time local updates of cached information.

### B.4 VISUALIZATION AND INTERACTION OF THE SIMULATOR

We developed a visual interface for the simulator using Unity3D. The simulator supports both online and offline demonstration capabilities. In offline demonstration mode, historical records of container entry and slot selection can be replayed to analyze storage effectiveness and algorithm performance. In online demonstration mode, the visual interface synchronizes with the execution progress of the algorithm, allowing users to interactively control the progress of container loading and storage through the frontend interface.

## C DATASETS

The dataset used in the simulator was extracted from the real historical data of Meishan Terminal in Ningbo-Zhoushan Port's Terminal Operating System (TOS) from October 2022 to December 2022. We selected a large-scale export container route from this period. To adapt to the simulator

environment, we excluded special containers that require separate storage areas, such as dangerous goods and refrigerated storage containers. For information security purposes, we retained only certain attributes of the containers, including container number, flow direction, route, size, discharge port, weight, status, storage position, entry time, and exit time. Based on the actual yard division criteria, we categorized the containers into 12 weight levels.

## D  IMPLEMENTATION OF BASELINE

The experiment was conducted in PyTorch. The hardware we used was a Windows 10 machine with a 12th Gen Intel(R) Core(TM) i5-12490F (@4.6GHz) CPU and an NVIDIA GeForce RTX 3060 GPU. The SAC models for the high-level agent and lower-level agent both consisted of a single embedding layer, an LSTM network based on the pointer network, an Attention module, and a mask module. The discount factor was set to 0.99, and the target entropy was set to -1 for both agents. The learning rate for SAC was set to 0.0001 for the actor, critic, and $\alpha$ in the higher level, and 0.00005 for all in the lower level. We used the Adam optimizer for all the optimizers used in our experiments.

## E  EXPERIMENT RESULT

### E.1  SUPPLEMENTAL EXPERIMENTAL RESULTS

| Experiment | Method | Block-equilibrium ↓ | Block Num ↓ | Reshuffle rate ↓ | Concentration ↓ | Yard-berth distance ↑ | Objective ↑ |
|---|---|---|---|---|---|---|---|
| 300-1 | Expert Strategy1 | 0.2452 | 0 | 0.1772 | -0.0498 | 1.0 | 12.7313 |
| | Expert Strategy2 | 0.2787 | 0 | 1.5518 | **-0.0598** | 1.0 | -127.0680 |
| | Expert Strategy3 | **0.0060** | 5 | 1.4481 | -0.0331 | 0.9753 | -143.3403 |
| | Random | 0.3250 | 5 | 0.1003 | 0.1225 | 0.9824 | -55.6716 |
| | **Hierarchical-SAC(Ours)** | 0.0423 | **0** | **0.0969** | -0.0381 | **1.0** | **39.8773** |
| 300-2 | Expert Strategy1 | 0.2976 | 0 | 0.1739 | -0.0548 | 1.0 | 8.3277 |
| | Expert Strategy2 | 0.2787 | 0 | 1.4816 | **-0.0598** | 1.0 | -120.0445 |
| | Expert Strategy3 | **0.0060** | 5 | 1.7491 | -0.0381 | 0.9753 | -172.9389 |
| | Random | 0.2307 | 5 | 0.1137 | 0.1197 | 0.9811 | -47.3620 |
| | **Hierarchical-SAC(Ours)** | 0.0423 | **0** | **0.0668** | -0.0397 | **1.0** | **43.0546** |
| 572-1 | Expert Strategy1 | 0.0369 | 0 | 0.1273 | -0.0328 | 1.0 | 36.8411 |
| | Expert Strategy2 | 0.5933 | 0 | 1.6945 | **-0.0650** | 0.875 | -178.5391 |
| | Expert Strategy3 | **0.0032** | 5 | 1.7382 | -0.0368 | 0.9709 | -171.9125 |
| | Random | 0.3482 | 5 | 0.1204 | 0.1114 | 0.9725 | -59.3828 |
| | **Hierarchical-SAC(Ours)** | 0.0401 | **0** | **0.0418** | -0.0171 | **1.0** | **43.5078** |
| 572-2 | Expert Strategy1 | 0.2976 | 0 | 0.1739 | -0.0548 | 1.0 | 8.3277 |
| | Expert Strategy2 | 0.5933 | 0 | 2.1937 | **-0.0650** | 0.875 | -228.4518 |
| | Expert Strategy3 | **0.0032** | 5 | 1.6736 | -0.0333 | 0.9709 | -165.8043 |
| | Random | 0.3584 | 5 | 0.1221 | 0.1042 | 0.9757 | -59.6955 |
| | **Hierarchical-SAC(Ours)** | 0.0423 | **0** | **0.0668** | -0.0397 | **1.0** | **43.0546** |
| 1000-1 | Expert Strategy1 | 0.0384 | 0 | 0.1689 | -0.0469 | **1.0** | 33.9667 |
| | Expert Strategy2 | 0.6298 | 0 | 2.2920 | **-0.0685** | 0.875 | -241.5824 |
| | Expert Strategy3 | **0.0081** | 6 | 1.6325 | -0.0460 | 0.9375 | -172.5890 |
| | Random | 0.4734 | 6 | 0.1859 | 0.0669 | 0.9512 | -85.0726 |
| | **Hierarchical-SAC(Ours)** | 0.0619 | **0** | **0.0979** | -0.0131 | 0.9780 | **34.2254** |
| 1000-2 | Expert Strategy1 | 0.0476 | 0 | 0.1859 | -0.0456 | **1.0** | 31.2039 |
| | Expert Strategy2 | 0.6298 | 0 | 2.3764 | **-0.0685** | 0.875 | -250.0280 |
| | Expert Strategy3 | **0.0081** | 6 | 1.7151 | -0.0446 | 0.9375 | -180.9898 |
| | Random | 0.4266 | 6 | 0.1734 | 0.0717 | 0.9521 | -79.5739 |
| | **Hierarchical-SAC(Ours)** | 0.0614 | **0** | **0.0817** | -0.0130 | 0.9780 | **35.8906** |

Table 5: Complete results of different methods on test datasets

### E.2  VISUALIZATION OF RESULTS

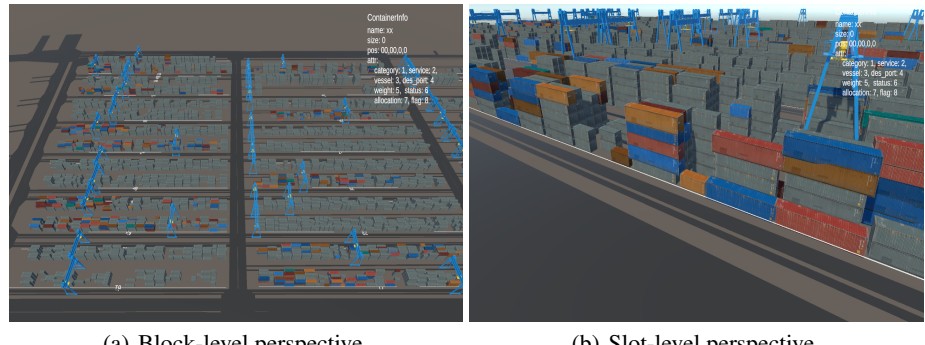

| (a) Block-level perspective | (b) Slot-level perspective |

Figure 6: Visualization results of dataset 1000 (Hierarchical-SAC)

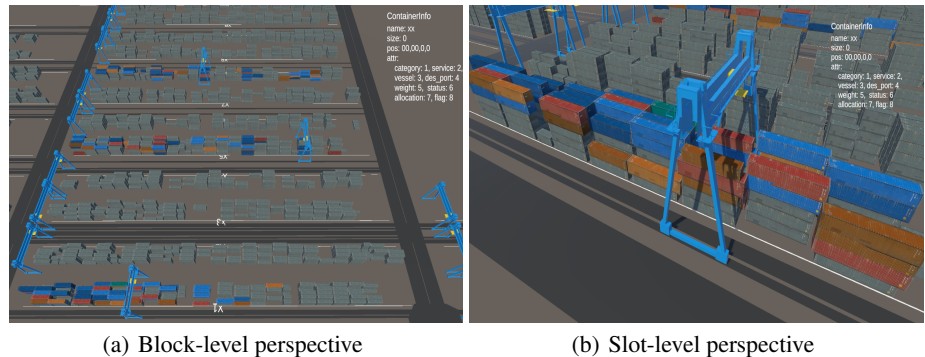

| (a) Block-level perspective | (b) Slot-level perspective |

Figure 7: Visualization results of dataset 300 (Hierarchical-SAC)

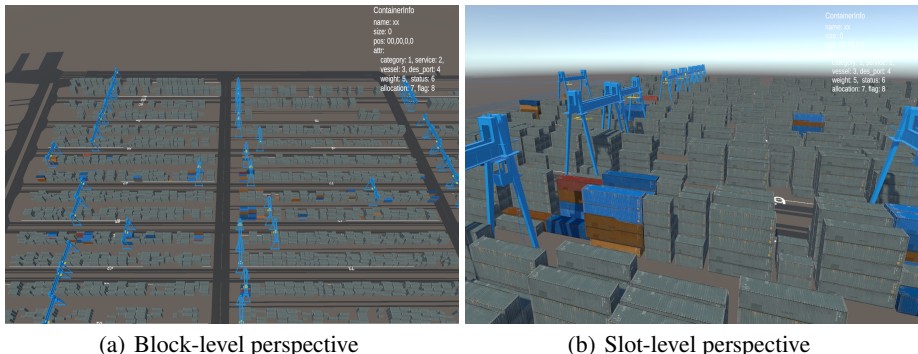

| (a) Block-level perspective | (b) Slot-level perspective |

Figure 8: Visualization results of dataset 572 (Expert Strategy2)

