# OpenReview forum: "A Hierarchical Reinforcement Learning Based Optimization FrameWork for Large Scale Storage Location Assignment Problem"
_ICLR.cc/2024/Conference — ICLR 2024 Conference Withdrawn Submission_

### Official Review · Reviewer_Kpm8 · 2023-10-24

**Soundness:** 1 poor
**Presentation:** 2 fair
**Contribution:** 2 fair
**Rating:** 3
**Confidence:** 4

**Summary:**

The paper describes a solution for storing containers in large container ports organized in blocks, bays, rows, and tiers. The agent is hierarchically organized, with a higher-level agent deciding on the block and a lower-level deciding the other three variables for storing a container. The reward function seems to be a 5-tuple of cost functions with are combined by a weighted sum. However, these seemed to be shaped rewards, and it is unclear if the final agent is evaluated by these shaped immediate rewards or something else.
The policy function is based on a pointer network with LSTM encoder and decoder and applied on both levels. The agent is trained with Soft Actor Critic.
The authors compare their agent with a flat SAC variant, a random agent, and three heuristics. The agents are compared with respect to the defined reward functions which makes it somewhat difficult to understand what the results should actually tell the reader.

**Strengths:**

The storage location assignment problem is exciting and seems to be quite challenging. Furthermore, the authors seem to be able to build a simulation environment based on real-world data from a large container port.
In general, using a hierarchical policy makes theoretically much sense and seems to be demonstrated from the experiments as well.

**Weaknesses:**

The formulation of the SLAP task is not exact enough to understand how the agent is trained and applied. In particular, it is unclear what happens in a step of the agent. Also, I am wondering what is the long-term goal of the agent. In particular, all named reward functions seem to be self-defined to offer immediate rewards to simplify training. However, this raises the question of whether directly trying to optimize a multi-armed bandit solution might not make more sense.

The technical contribution to Deep Reinforcement learning does not seem very broad. Applying the hierarchical decision paradigm seems straightforward. The proposed policy function is not exactly new, and applying SAC is also out of the box.

The paper doesn't do an good job of being reproducible. A lot of statements are unclear or only explained in the appendix. Some references are hard to follow (e.g., where is 4.3.1 A ?).

To conclude, the paper does not provide enough details and structure to be reproducible. Furthermore, the technical contribution seems limited to a rather straight-forward build.

**Questions:**

What is the long-term goal of the agent?

The state description describes the storage facilities' remaining capacities, but the simulation's temporal flow is unclear. Is every container handled one after the other? Is there a schedule for emptying the storage facilities due to leaving vessels?
It is kind of hard to understand for a non-expert how the agent is supposed to learn the mechanics of the environment as it seems to mostly encode the current capacity.

---

### Official Review · Reviewer_JQZq · 2023-10-31

**Soundness:** 3 good
**Presentation:** 2 fair
**Contribution:** 2 fair
**Rating:** 3
**Confidence:** 4

**Summary:**

In this paper, a hierarchical optimization framework based on RL is proposed to addresses the Storage Location Assignment Problem (SLAP), that is an essential problem within the domain of logistics. A container storage allocation simulator based on real port data is established to evaluate the performance of proposed method. As shown in the experimental results, the proposed hierarchical reinforcement learning method outperforms the baseline methods under various settings.

**Strengths:**

1. The language is mostly easy to follow.
2. This paper introduces the hierarchical reinforcement learning into a real-world application scenario, i.e., the storage location assignment problem.
3. The proposed simulator simulates an actual industrial scenario in the real world, and the data is collected from terminal operating system of Ningbo-Zhoushan Port. Therefore, the work has certain potential to guide actual industrial production.
4. The proposed method does exhibit some performance improvements over simple baselines.

**Weaknesses:**

1. The technical novelty is limited. Hierarchical reinforcement learning is known to be able to deal with large action space, and has been applied to combinatorial optimization (e.g., [Ma2019]). The neural architecture is quite standard in neural combinatorial optimization, which has already been explored in maritime applications (e.g., [Jin2023]).

2. The baselines are too weak. The "expert strategies" are very simple heursitic rules. For expert strategy 1, directly selecting actions by maximizing immediate reward is known to be a poor policy for sequential decision making. For other strategies, there is no evidence that they will optimize the complicated reward defined in this paper. More advanced SLAP methods (which are available in the literature) should be compared against.

3. This paper lacks a comprehensive problem description, and many details are unclear. In particular, it is unclear how to weight the five terms in the objective function (which also serves as a reward).

4. This article is not well organized, as much of the valuable information placed in the appendices.

[Ma2019] Ma, Qiang, et al. "Combinatorial optimization by graph pointer networks and hierarchical reinforcement learning." arXiv preprint arXiv:1911.04936 (2019).
[Jin2022] Jin, Xin, et al. "Container stacking optimization based on Deep Reinforcement Learning." Engineering Applications of Artificial Intelligence 123 (2023): 106508.

**Questions:**

1. Please add a detailed description of the objective function.
2. Please check and add missing notation explanation, such as the notations in Equation 1.
3. Some abbreviations lacking explanation (e.g., LLA and HLA in figure.2) make it difficult to follow.
4. To illustrate the innovative nature of the proposed methodology, the novel aspects of the proposed method and why it is well-suited for the SLAP should be extensively discussed.
5. In order to demonstrate the superiority of the proposed methodology, some baselines with better performance should be added, such as meta-heuristic methods and DRL methods.
6. The DRL baseline, single SAC, cannot learn any effective policies. Is this phenomenon caused by hyper-parameters? Some DRL algorithms that can learn effective policies for SLAP should be included as baselines.
7. The computational complexity should be rigorously discussed in this paper.

---

### Official Review · Reviewer_Bh6D · 2023-10-31

**Soundness:** 2 fair
**Presentation:** 2 fair
**Contribution:** 2 fair
**Rating:** 3
**Confidence:** 4

**Summary:**

The study presents a two-layers method for large-scale storage location allocation. An upper-level agent selects blocks, and a lower-level agent chooses specific slots. Both use LSTM and attention-based networks trained via the Soft Actor-Critic algorithm. The approach is validated using a simulation built from real data.

**Strengths:**

1.The paper is generally clear.
2.Research into the large-scale storage location assignment problem is meaningful.

**Weaknesses:**

1. The article seems to lack originality, as using a hierarchical framework to tackle complex, large-scale problems is a intuitive approach. The neural network architecture adopted, similar to pointer network, is a typical model within the ML4CO field.
2. Despite the introduction mentioning that "numerous studies have utilized traditional operations research (OR) or metaheuristic methods, such as intelligent optimization algorithms, to address SLAPs", the experiment section's baselines are predominantly several manually-designed priority rules.
3. The experimental results indicate that the proposed hierarchical, learning-based algorithm does not outperform all the baselines across every metric, and the improvement over Expert Strategy1, in terms of Objective, is not markedly significant.

**Questions:**

1. Is it possible to introduce some baselines based on operations research or metaheuristic methods?
2. The authors included a link in the contribution section of the introduction that directs to a non-existent/private GitHub repository. I'm not sure if this is appropriate under double-blind review.

---

### Official Review · Reviewer_Bseu · 2023-11-01

**Soundness:** 2 fair
**Presentation:** 3 good
**Contribution:** 2 fair
**Rating:** 5
**Confidence:** 4

**Summary:**

This paper proposes a learning-based approach to solve the large-scale Storage Location Assignment Problem (SLAP). Specifically, the proposed framework is a two-level model that learns to determine which block to choose and select the final storage location under the constraints of the selected blocks. The paper also introduces a policy network based on attention mechanisms to improve the performance of the proposed framework. The experiments conducted on two different scales of export container route data demonstrate the effectiveness of the proposed approach.

**Strengths:**

1. The paper proposes a novel approach to solve large-scale Storage Location Assignment Problem using hierarchical reinforcement learning. The problem setting is rarely studied and difficult.
2. The proposed framework is a two-level model that learns to determine which block to choose and select the final storage location under the constraints of the selected blocks. This approach is more efficient and effective than existing methods.
3. The paper uses real historical data from the terminal operating system of Ningbo-Zhoushan Port to validate the proposed framework, which makes the results more reliable and applicable to real-world scenarios.
4. The paper conducts extensive offline simulations and online testing using a realistic container terminal simulator based on real data to validate the superior performance of the proposed framework compared to existing benchmark methods.

**Weaknesses:**

1. The paper does not compare the proposed framework with other state-of-the-art (deep) reinforcement learning methods, which limits the ability to assess the effectiveness of the proposed approach. The baselines seem to be too basic.
2. The designed deep policy network is not novel.
3. In Table I, computation time of all methods are expected.

**Questions:**

1. Please show more theoretical analysis on computational complexity mentioned in Contribution.
2. How does the proposed framework compare to existing methods for solving the Storage Location Assignment Problem?
3. How does the proposed framework handle the uncertainty of incoming containers and the real-time nature of the problem?
4. How can the proposed framework be extended to other related problems in logistics and supply chain management?
5. Is the proposed framework applicable to other ports or logistics scenarios, or is it specific to the dataset from Ningbo-Zhoushan Port? Could you list more results on different dataset, including randomly generated ones?
6. How does the training time of the proposed framework, especially compared to other methods?